# A Small-Angle Neutron Scattering, Calorimetry and Densitometry Study to Detect Phase Boundaries and Nanoscale Domain Structure in a Binary Lipid Mixture

**DOI:** 10.3390/membranes13030323

**Published:** 2023-03-10

**Authors:** Natalie Krzyzanowski, Lionel Porcar, Ursula Perez-Salas

**Affiliations:** 1Department of Physics, University of Illinois at Chicago, Chicago, IL 60608, USA; 2Large Scale Structures Group, Institut Laue-Langevin, CEDEX 9, 38042 Grenoble, France

**Keywords:** lipid phase separation, binary lipid phase diagram, dipalmitoylphosphocholine, DPPC, dilauroylphosphocholine, DLPC, unilamellar vesicles, model membrane, small-angle neutron scattering, contrast matching, calorimetry, densitometry, MONSA, bead vesicle model

## Abstract

Techniques that can probe nanometer length scales, such as small-angle neutron scattering (SANS), have become increasingly popular to detect phase separation in membranes. But to extract the phase composition and domain structure from the SANS traces, complementary information is needed. Here, we present a SANS, calorimetry and densitometry study of a mixture of two saturated lipids that exhibits solidus–liquidus phase coexistence: 1,2-dipalmitoyl-d62-*sn*-glycero-3-phosphocholine (dDPPC, tail-deuterated DPPC) and 1,2-dilauroyl-*sn*-glycero-3-phosphocholine (DLPC). With calorimetry, we investigated the phase diagram for this system and found that the boundary traces for both multilamellar vesicles (MLVs) as well as 50 nm unilamellar vesicles overlap. Because the solidus boundary was mostly inaccessible by calorimetry, we investigated it by both SANS and molecular volume measurements for a 1:1 dDPPC:DLPC lipid mixture. From the temperature behavior of the molecular volume for the 1:1 dDPPC:DLPC mixture, as well as the individual molecular volume of each lipid species, we inferred that the liquidus phase consists of only fluid-state lipids while the solidus phase consists of lipids that are in gel-like states. Using this solidus–liquidus phase model, the SANS data were analyzed with an unrestricted shape model analysis software: MONSA. The resulting fits show irregular domains with dendrite-like features as those previously observed on giant unilamellar vesicles (GUVs). The surface pair correlation function describes a characteristic domain size for the minority phase that decreases with temperature, a behavior found to be consistent with a concomitant decrease in membrane mismatch between the liquidus and solidus phases.

## 1. Introduction

The major constituents of biological membranes are lipids. Because lipids are amphiphilic, they spontaneously self-assemble into the essential feature of biological membranes: a lipid bilayer. Lipid bilayers thus serve as models for biological membranes and have been the subject of intense research for over half a century. Early on, it was recognized that the lipid composition in biological membranes is immensely diverse—a critical feature for function [1]. This led to the study of relatively simple lipid mixtures, such as binary lipid mixtures. Although simple, even binary mixtures produce somewhat sophisticated compositional phase diagrams. The intent of understanding the behavior of simple lipid mixtures was that they could suggest possible features of biological membranes as well [2]. Indeed, a thermodynamic framework led to a deeper understanding of these mixtures, from molecular cooperativity to non-ideal mixing characteristics [3,4,5,6,7], which became the precursor of what later became the lipid raft hypothesis [8,9,10].

The lipid raft hypothesis proposes that rather than having biological membrane constituents be randomly mixed, the cell membrane partitions lipids, sterols and proteins into laterally segregated regions. Although some cell systems have been found to exhibit observable micron-scale domains [11,12], the majority of cell membranes do not, which led to the idea that lipid rafts were still present but were mostly small (nanometers), dynamic and possibly transient [13]. Thus, techniques that study membranes at the nanoscale have been heavily pursued. Small-angle neutron scattering (SANS) stands out as a technique to probe this length scale because even though it is considered to be a low-resolution structure technique, with the use of selective deuteration, it has exquisite sensitivity to the emergence of nanoscale heterogeneities in membranes [14] and, as a result, has helped advance the field both using membrane mimics [15] as well as living cells [16]. In addition, analysis of the structures formed by phase separation in model membranes has given insights into, for example, the role of membrane mismatch between domains and the surrounding membrane environment and the size of the domains [17]. 

Analysis of SANS data from membranes exhibiting domains has mostly been conducted using analytical models with predefined shapes, such as stripes and circular domains [18,19,20]. More recently, Bolmatov et al. developed a promising scattering model that uses a phenomenological free energy approach and whose fitting parameters could in principle identify domain configurations having diverse shapes; however, its implementation is still ongoing [21]. Indeed, as shown by Heberle et al. [15], the structures of phase-separated membranes can be quite complex. One example is the binary mixture of two saturated phosphatidylcholine lipids having different tail lengths: 16- and 12-length carbons (di-pamitoyl phosphatidylcholine DPPC and di-lauroyl phosphatidylcholine DLPC, respectively), which does not display round domains or simple stripes [22,23]. Indeed, limiting the analysis of SANS data from membranes exhibiting complex structures to these analytical shapes can produce inconsistencies with other aspects of these membranes. For example, Anghel et al. [18], who studied a deuterated analog for this system (dDPPC:DLPC, where dDPPC differs from DPPC by having deuterated tails) by SANS, obtained fits that suggested compositions that conflicted with the known phase diagram for DPPC:DLPC reported by Van Dijck et al., obtained by calorimetry [3]. 

In this work, we explore the analysis of SANS spectra from 1:1 dDPPC:DLPC vesicles using an alternative strategy: bead vesicles. The roughly 1200 beads forming the vesicle provide for local variations in composition and collectively are not constrained to any particular shape. Using this approach, the resulting membrane structures or patterns found were similar to those observed in large unilamellar vesicles by microscopy [22,23]. The resulting irregular domains in the bead vesicles could be characterized by a surface bead pair correlation function, from which a temperature-dependent characteristic length was obtained. 

The resulting phase diagram for the binary mixture of dDPPC and DLPC, which combined calorimetry, densitometry and SANS analysis, produced the elusive solidus boundary for the system. Together, the solidus and liquidus boundaries describe a non-ideal system that follows the thermodynamics of regular solution theory [3]. Interestingly, the combined approach of calorimetry, densitometry and SANS revealed details of the behavior of these lipids in each phase—particularly the volume of DLPC in the solidus phase—which had not been reported before. 

## 2. Materials and Methods

### 2.1. Materials

1,2-dipalmitoyl-*sn*-glycero-3-phosphocholine (hDPPC), 1,2-dipalmitoyl-d62-*sn*-glycero-3-phosphocholine (dDPPC) and 1,2-dilauroyl-*sn*-glycero-3-phosphocholine (DLPC) were purchased from Avanti Polar Lipids (Alabaster, AL, USA) as lyophilized powders and stored at −80 °C until further use. 

### 2.2. Differential Scanning Calorimetry (DSC)

Using a micro differential scanning calorimeter, µDSC III from SETARAM Instrumentation, we obtained the phase diagram for dDPPC:DLPC. DSC data were taken for both 50 nm unilamellar vesicles (SUVs) and multilamellar vesicles (MLVs). dDPPC and DLPC in powder form were weighed into a glass vial and thoroughly mixed using chloroform (vortexed vigorously until the solution was clear). The removal of this chloroform was performed first with a stream of N_2_ gas using a nitrogen evaporation system (N-EVAP) until a thin in-appearance dry lipid film remained on the vial’s walls. The vials were then placed in vacuum at 60 °C overnight to remove any residual chloroform. Once dry, the sample was hydrated with 1 ml of H_2_O. MLVs were prepared by leaving the solution on a shaker overnight mixing in a 47 °C oven until the resulting mixture became a homogenous suspension. The SUVs were prepared, subsequently, by extrusion. The extrusion was performed with a modified Avanti Polar Lipids extruder system that included heating via an Anova water bath (see Appendix A). A syringe pump mechanically manipulated the syringes of the extruder (New Era Pump Systems Inc., Farmingdale, NY, USA). The final extrusion consisted of 41 passes through a polycarbonate filter with a 50 nm average pore size. The extrusion of the vesicles was performed at 50 °C, which is well above the melting temperature of the lipid system. The speed of extrusion can be controlled on the syringe pump, which allows extruding of even relatively high-concentration solutions. 

To obtain the phase diagram of dDPPC:DLPC, thermograms for several dDPPC:DLPC ratios were collected. Keeping DLPC constant (15 mg/mL), the amount of dDPPC was varied from 0 mol% to 90 mol%. The 100 mol% dDPPC was run at 15 mg/mL. SUVs were obtained from the MLV solutions after their calorimetry data were collected. Thermogram scans were obtained at 0.2 °C/min. SetSoft 2000 provided by SETARAM was used to collect DSC data and exported as a text file to use with the program Mathematica™ for baseline subtraction and determination of onset and completion temperatures (details in Appendix A of the Appendix A). 

### 2.3. Densitometry

Densitometry data were obtained on samples having a total 10 mg/mL lipid concentration using the Density and Sound Velocity Meter: DMA 5000 M from Anton-Paar (Ashland, Virginia) at the Partnership for Soft Condensed Matter (PSCM) in Grenoble, France. Three samples were measured by cooling: dDPPC, DLPC and 1:1 dDPPC:DLPC. The lipid samples consisted of multilamellar vesicles (MLVs) in H_2_O, which had been incubated and gently rocked overnight at 40 °C. The samples were loaded at room temperature, and densitometry data were taken from 50 °C down to 5 °C in steps of 1 °C. 

### 2.4. Small-Angle Neutron Scattering

SANS measurements were taken on 50 nm SUV vesicle solutions using a 1:1 ratio of dDPPC:DLPC. The measurements were performed on the D22 instrument at Institut Laue-Langevin (Grenoble, France). Instrument configuration covered a Q range of 0.003 ≤ Q ≤ 0.5 Å^−1^. The wavelength used was 6 Å with a wavelength resolution of Δλ/λ = 10%. Data were collected with a 2-D detector and reduced using GRASP, the reduction package provided by the ILL or the reduction package provided by NCNR in IGOR [24]. 

## 3. Results

SANS is a powerful technique to obtain the structural information of particles that range from a few to tens of nanometers in size [25]. This is because the scattered intensity, $I\left(Q\right)$, is directly related to the particle’s shape, size and composition. For a dilute, monodisperse particle solution, $I\left(Q\right)$ is given by: (1)$$I\left(Q\right)-I_{incoh}=vV_{particle}{\left(SLD_{particle}-SLD_{solvent}\right)}^{2}P{\left(Q\right)}_{particle}$$
where $P\left(Q\right)$ is the form factor of the particles and contains information about their shape and size. v is the volume fraction of particles, and $V_{particle}$ is the volume of the particle. The scattering length density (SLD) is a measure of the interaction of neutrons with the atomic and isotopic species in the particles’ and solvent’s chemical composition. As shown in Equation (1), to be able to extract information about the particles’ shape and size through $P\left(Q\right)$, contrast is key, i.e., ${\left(SLD_{particle}-SLD_{solvent}\right)}^{2}>0$. 

Vesicles whose membranes are composed of a 1:1 molar mixture of DPPC and DLPC will phase separate and form lipid domains—solidus and liquidus phases—with distinct lipid compositions given by the phase diagram [3]. The approach to achieve contrast between these domains and be measured by SANS is to substitute one of the lipids with its deuterated version. This approach does not change the chemical identity of the lipid, but it does change its SLD significantly. For example, at room temperature, the tail SLD of DPPC (in its fully hydrogenated form) is −0.4 × 10^−6^ Å^−2^, while for dDPPC (with all 62 tail hydrogens substituted with deuteriums), it is 7.5 × 10^−6^ Å^−2^. Hence, when a mixture consisting of a deuterated lipid species (dDPPC) and a hydrogenated lipid species (DLPC) phase separates, the phase-separated regions acquire different SLDs because of the difference in lipid composition in each phase. To drastically highlight these differences in SLD between phases in the membrane, the solvent’s SLD is chosen to match the membrane’s mean SLD, typically obtained when the membrane’s lipids are fully mixed—in a single phase—above the miscibility transition [14]. At this contrast match condition (when $SLD_{vesicle}=SLD_{solvent}$), the scattering becomes flat (equal to the background) and featureless (as shown by Equation (1)). However, when the temperature is lowered below the miscibility temperature, phase separation emerges and, because the SLDs of the domains do not match each other or that of the solvent, scattering emerges too. Figure 1 shows the scattering from 1:1 dDPPC:DLPC 50 nm in diameter vesicles in their contrast-matched (CM) solvent (see Appendix A in the Appendix A for details on obtaining the CM condition for the system) above and below the miscibility temperature. At 40.6 °C, which is above the miscibility transition for this mixture, the scattering signal is flat, confirming that the membrane’s SLD has been matched to the solvent’s SLD. Upon lowering the temperature, below the miscibility transition, phase separation into solidus and liquidus domains ensues, and their presence is clearly captured by a significant increase in the scattering intensity [14]. This emergent scattering, however, is not due to the vesicle form factor as it is contrast-matched to the solvent but due to the characteristics of the domains in the vesicles. Extracting structure and composition information from these coexisting phases in vesicles, beyond just detecting their presence, requires that the collected scattering signal captures the peak resulting from the size of the domains present in the vesicles. Using 50 nm vesicles allows the capture of this peak [17,21], as it is clearly apparent at 24 °C and 22 °C. As temperature decreases, we observe the scattering signal evolve. Initially, the scattering peak increases in intensity and shifts to higher Q values (from 24 °C to 22 °C), and then low Q scattering increases while the scattering peak lowers in intensity while still shifting to higher Q values. At the lowest temperatures, the scattering curves have significant low Q scattering while the peak has become broad and shoulder-like in the high Q. This evolution in the scattering spectra suggests that the membranes’ domains are not only changing in lipid composition, size and number but that the lipids’ SLD is changing as well. Therefore, in order to quantitatively analyze the SANS data presented in Figure 1, we need to constrain the analysis with the thermodynamic parameters of the system. The phase diagram provides the lipid composition of each phase and fraction of each phase [26] while the volumes of the lipids provide the SLD of each phase [27].

The phase diagram for DPPC:DLPC has been previously reported [3,28]; however, because of our use of the tail-deuterated DPPC (dDPPC), we revise it. Calorimetry is a technique that is commonly used to generate the phase diagram from heat capacity traces. Figure 2A shows the specific heat capacity (cooling) traces for dDPPC:DLPC compositions varying from 100 mol% dDPPC to 100 mol% DLPC. The phase diagram’s liquidus and solidus boundaries are determined by the onset and completion temperatures of phase transitions (see Appendix A for details; Appendix A). The onset and completion temperatures of the phase transitions detected in the heat capacity trace of a given mixture are marked with dots in Figure 2A and displayed as a composition versus temperature phase diagram in Figure 2B. Since the SANS experiments were performed on 50 nm in diameter small unilamellar vesicles (SUVs), Figure 2B also shows the position of the onset and completion temperatures for both multilamellar vesicles (MLVs) and 50 nm SUVs (see Appendix A showing the heat capacity traces for MLVs during heating and for SUVs during cooling). As observed in Figure 2B, the boundaries of the phase diagram remain the same for both MLVs and SUVs, demonstrating that the high membrane curvature in SUVs does not affect the phase diagram boundaries of the system. 

The phase diagram for dDPPC: DLPC, shown in Figure 2B, indicates that the system exhibiting a single phase is at a high temperature, but when the temperature drops and crosses the upper miscibility boundary, the system separates into a solidus and a liquidus phase. The molecular composition of each phase as well as the overall fraction of each phase at any given temperature (through the lever rule—see more details in Appendix A of the Appendix A) is extracted from the upper miscibility or liquidus boundary and the lower miscibility or solidus boundary of the phase diagram. The calorimetry phase diagram shown in Figure 2B describes a well-defined liquidus boundary for all dDPPC compositions; however, the solidus boundary appears to have a gap or discontinuity between *χ_dDPPC_* = 0.6 and *χ_dDPPC_* = 0.7. Indeed, a similar behavior for the solidus boundary was reported for the DPPC:DLPC system by Van Dijck et al. [3]. As we will show, by combining this calorimetry information with the required thermodynamic volumes of the lipids in the solidus and liquidus phases in the analysis of the SANS data, we will be able to retrieve the elusive solidus boundary for the system.

In order to obtain the thermodynamic molecular volumes of the lipids, we used densitometry. Our densitometry measurements consisted of measuring the density of a mixture of lipids in water, which is given by: (2a)$$\rho_{mixture}\left(T\right)=\frac{w_{lipids}+w_{water}}{V_{lipids}+V_{water}}$$
where $w_{lipids}$ and $w_{water}$ are the weights and $V_{lipids}$ and $V_{water}$ are the volumes of lipids and water, respectively. From Equation (2a), we can solve for the molecular volume of the lipids, $V_{L}$: (2b)$$V_{L}\left(T\right)=\frac{M_{w\;}\times 10^{24}{}_{\;}}{N_{A}}\frac{1}{w_{lipids}}\left(\frac{w_{lipids}+w_{water}}{\rho_{mixture}\left(T\right)}-\frac{w_{water}}{\rho_{water}\left(T\right)}\right)$$

Here, $N_{A}$ is Avogadro’s number and $M_{w}$ is the molecular weight of the lipid system. The factor $10^{24}$ is a unit conversation factor from cm^3^ to ${Å}^{3}$, since the density was obtained in g·cm^−3^. Because we performed the density measurements in H_2_O, we also measured $\rho_{H2O}$ separately (see Appendix A). Figure 3 shows a plot of the molecular volume, $V_{L}$, for DLPC, dDPPC and the 1:1 dDPPC:DLPC mixture as a function of temperature in the range between 5 °C and 50 °C. We observe the expected sharp molecular volume change in dDPPC between the fluid and the gel state at the known *T_m_* (melting or transition temperature) value of 37 °C [29]. Below and above 37 °C, the molecular volume change in dDPPC is linear and slowly varies with temperature. DLPC’s molecular volume, which is in the fluid phase between 5 °C and 50 °C, is also observed to be linear and slowly varying with temperature. Interestingly, away from the *T_m_*, the change in the molecular volumes for both dDPPC and DLPC is very similar, both displaying similar slopes (see Appendix A). In addition, as shown in Appendix A, we find that these molecular volumes are consistent with those previously reported for dDPPC [30,31,32,33] and DLPC [33,34]. 

For the 1:1 dDPPC:DLPC mixture, we also observe that the molecular volume behaves linearly above the miscibility temperature (*T* > 27 °C) with a slope consistent with those of dDPPC and DLPC in the fluid phase. However, for temperatures below its miscibility temperature (≈27 °C), the volume decreases more rapidly and nonlinearly. In this region, the system exhibits the coexistence of solidus and liquidus phases. As a reference, the molecular volume average of dDPPC and DLPC in the 5 °C to 50 °C range is shown with a black trace in Figure 3. We find that above 37 °C, this molecular volume average trace for dDPPC and DLPC exactly coincides with the molecular volume behavior of the 1:1 mixture. This linear behavior, however, extends to ≈ 27 °C and thus differs from the molecular volume average of dDPPC and DLPC, which displays a sharp drop at 37 °C. Starting at ≈27 °C, the molecular volume for the 1:1 mixture decreases smoothly but nonlinearly as the temperature drops. Notably, below 18 °C, the molecular volume average of dDPPC and DLPC becomes larger than the measured molecular volume for the mixture. This behavior revealed that a fraction of DLPC must be in the gel state. Hence, we inferred that the solidus phase must only comprise lipids in the gel state while the liquidus phase must only consist of lipids in the fluid state. Indeed, from the molecular volume behavior of the 1:1 mixture, we find that dDPPC remains fluid well below its *T_m_*, down to ≈27 °C, and will remain fluid while in the liquidus phase, while DLPC, which is fluid above freezing temperatures, will adopt a gel-state volume in the solidus phase up to ≈27 °C. However, what are the fluid molecular volume of dDPPC below 37 °C and the gel-state volume of DLPC above −5 °C? We assume that the fluid molecular volume of dDPPC below 37 °C is a linear extrapolation of its fluid molecular, and similarly, the gel-state volume of DLPC above −5 °C must be a linear extrapolation of the gel-state molecular volume of DLPC [27]. Support for this assumption comes from the observation that the extrapolated fluid molecular volume of dDPPC averaged with the fluid molecular volume of DLPC reproduced the molecular volume behavior of the 1:1 mixture above 27 °C. 

In order to obtain the gel-phase molecular volume for DLPC, which could not be obtained by the densitometry approach because the lowest measurable temperature was 5 °C, we instead used the calorimetry traces for which lower temperatures were recorded. Indeed, as demonstrated by Ebel at al. [35], molecular volumes can be derived from heat capacity traces as follows:(3)$$V_{L}\left(T\right)=-\frac{M_{w\;}\times 10^{24}{}_{\;}}{N_{A}}\int_{T}^{T_{max}}\left(\gamma c_{p}\left(T\right)+\alpha \right)\;dT+V_{L}\left(T_{max}\right)\;T<T_{max}$$
where α and γ are constants; α determines the linear behavior of the molecular volume away from the phase transition, while γ is a pre-factor controlling the magnitude of the volume change at the transition. 

Since the molecular volumes obtained from calorimetry had to coincide with those obtained by densitometry, the values for γ and α were varied until the inverted calorimetry curve overlapped with that obtained from densitometry. In all cases, $T_{max}$ was chosen to be well above the miscibility temperature. Figure 3 shows inverted calorimetry curves for dDPPC, DLPC and 1:1 dDPPC:DLPC, using Equation (3). The values for γ and α in each case are in Appendix A. As shown in the table, for dDPPC, γ was found to be $8.5\pm 0.5\times 10^{-4}$ cm^3^·J^−1^, which is within the reported value by Ebel et al. [35] ($8.5\pm 0.8\times 10^{-4}$ cm^3^·J^−1^). Ebel et al. also reported that this value of γ was independent of the lipid type; however, for the 1:1 dDPPC:DLPC mixture and for DLPC, we found that γ was slightly larger ($13.7\pm 0.6\times 10^{-4}$ cm^3^·J^−1^ and $15.2\pm 0.8\times 10^{-4}$ cm^3^·J^−1^, respectively). 

Because the molecular volumes obtained for DLPC from densitometry did not reach the region where DLPC transitions to the gel phase, the value of γ for DLPC had to be obtained following a self-consistency condition: at −5 °C, where DLPC and dDPPC are in the gel phase, the molecular volume for the 1:1 dDPPC:DLPC mixture had to be equal to the mean molecular volume of dDPPC and DLPC. Since molecular volumes from the inverted calorimetry curves for dDPPC and the 1:1 dDPPC:DLPC mixture were known at −5 °C, the molecular volume of DLPC in the gel phase was obtained and γ was determined. Figure 3 shows the inverted calorimetry curve for DLPC resulting from the condition that all lipids are in the gel phase at −5 °C. In the figure, the extrapolated molecular volumes for dDPPC in the fluid phase below 37 °C and DLPC in the gel state above −5 °C are marked with red arrows. Appendix A provides the linear equations that describe both the gel and fluid molecular volumes of dDPPC and DLPC for all temperatures. 

With this information, it is now possible to obtain the temperature-dependent molecular SLDs for dDPPC and DLPC: (4a)$${\mathrm{SLD}}_{dDPPC_{j}}=\frac{b_{dDPPC}}{V_{dDPPC_{j}}}$$
(4b)$${\mathrm{SLD}}_{DLPC_{j}}=\frac{b_{DLPC}}{V_{DLPC_{j}}}$$

The index *j* indicates gel (g) or fluid (f). b is the scattering length, which only depends on the atomic and isotopic make-up of, in this case, the lipid indicated in the subscript. The scattering lengths used in this work are listed in Appendix A.

Fitting of the SANS data requires the SLD of the liquidus and solidus domains given by:(5a)$${\mathrm{SLD}}_{s}=v_{dDPPC_{s}}{\;\mathrm{SLD}}_{dDPPC_{g}}+\left(1-v_{dDPPC_{s}}\;\right){\mathrm{SLD}}_{DLPC_{g}}$$
(5b)$${\mathrm{SLD}}_{l}=v_{dDPPC_{l}}{\;\mathrm{SLD}}_{dDPPC_{f}}+\left(1-v_{dDPPC_{l}}\;\right){\mathrm{SLD}}_{DLPC_{f}}$$
where the indexes *s* and *l* indicate solidus and liquidus, respectively, and g and f indicate gel and fluid, respectively. The terminology difference between solidus and gel or liquidus and fluid is meant to distinguish between the state of a lipid mixture versus the state of a lipid species. As will be shown below, this distinction is necessary.

Because there is a gap in the solidus boundary, the value of $v_{dDPPC_{s}}$ is not known. Fits to the SANS data required varying this parameter—or, equivalently, the value of ${\mathrm{SLD}}_{s}$—as will be detailed below. 

Since the temperature dependence of the molecular volumes of the lipids is critical information to properly calculate SLDs, we additionally considered the temperature-dependent SLD of the solvent. In the case of a mixture of H_2_O and D_2_O, the ${\mathrm{SLD}}_{solvent}$ is given by:(6)$${\mathrm{SLD}}_{solvent}=\psi \frac{b_{D2O}}{w_{D2O}/\rho_{D2O}}+\left(1-\psi \right)\frac{b_{H2O}}{w_{H2O}/\rho_{H2O}}$$
where ψ is the volume fraction of D_2_O in the mixture, $w_{D2O}$ and $w_{H2O}$ are the weights of H_2_O and D_2_O, respectively, used to make the solvent and $\rho_{D2O}$ and $\rho_{H2O}$ are the temperature-dependent densities for D_2_O and H_2_O, respectively [36] (see Appendix A).

Fitting of the SANS data also requires the SLD of the vesicles when the lipids are uniformly—ideally—mixed, i.e., in a single fluid phase. In this case, the vesicle’s mean SLD is given by:(7)$${\mathrm{SLD}}_{mean\;vesicle\;T>T_{m}}=v_{dDPPC_{f}}{\;\mathrm{SLD}}_{dDPPC_{f}}+\left(1-v_{dDPPC_{f}}\right){\mathrm{SLD}}_{DLPC_{f}}$$

Here the corresponding volume fraction of dDPPC in the system reduces to:(8)$$v_{dDPPC_{f}}=\frac{V_{dDPPC_{f}}\chi_{dDPPC}^{⋇}}{V_{dDPPC_{f}}\chi_{dDPPC}^{⋇}+V_{DLPC_{f}}\left(1-\chi_{dDPPC}^{⋇}\right)}$$
where $\chi_{dDPPC}^{⋇}$ corresponds to the molecular fraction of dDPPC in the system under study. For our SANS measurements, we chose $\chi_{dDPPC}^{⋇}$ = 0.5 

Since the main aim of the present work was to explore how SANS could be used to determine the structures that emerge from coexisting solidus and liquidus phases in membranes, we explored the analysis of SANS data using bead vesicles. Our bead vesicles were beads on a coordinate grid producing vesicles of a desired radius and membrane thickness and with the beads forming a near-single layer of beads representing the membrane (see Appendix A). The beads provide a mechanism for local variations in SLD and collectively are not constrained to any particular shape on the vesicle. The software we used to vary the beads’ SLD and fit the SANS data was MONSA [37]. MONSA calculates a scattering intensity from the real-space bead model; however, since extruded vesicles are polydisperse in size (~0.25), MONSA cannot properly generate the scattering of this system. Nevertheless, in this work, it is the presence domains that generate the excess scattering because the vesicles are basically contrast matched. As a result, the polydispersity in the vesicle size is insignificant and only the domains, having different shapes and sizes, contribute to the scattering. These varied domains can be represented in the bead vesicles and thus properly computed by MONSA. 

As shown in Figure 4A, we found that the bead vesicle scattering calculated with a single uniform SLD value (${\mathrm{SLD}}_{mean\;vesicle\;}$)—orange trace—or fitted with two SLD values—gray trace—corresponding to ${\mathrm{SLD}}_{dDPPC_{f}}$ and ${\mathrm{SLD}}_{DLPC_{f}}$, as described by Equation (7), properly describe the high-temperature contrast-matched vesicles at 40.6 °C. The fit—where the beads are allowed to take either value (${\mathrm{SLD}}_{dDPPC_{f}}$ or ${\mathrm{SLD}}_{DLPC_{f}}$,)—produced not only the expected molecular volume fraction of dDPPC (Equation (8)) in the 1:1 dDPPC:DLPC mixture of 55.3 ± 0.1 vol% but also showed a random distribution of the beads with the two SLDs as shown in Figure 5A. 

For all scattering data in the two-phase coexistence region (T≤ 24 °C), the beads were allowed to take either a liquidus or a solidus SLD value: ${\mathrm{SLD}}_{s}$ or ${\mathrm{SLD}}_{l}$ (Equation (5)). The value for ${\mathrm{SLD}}_{l}$ was directly obtained from the well-defined liquidus boundary of the phase diagram (Figure 2B and Appendix A); however, because the phase diagram shows a discontinuity in the solidus boundary in the temperature range probed by SANS, the value of ${\mathrm{SLD}}_{s}$ was varied by varying the ratio of dDPPC to DLPC in this phase (i.e., by varying the $f_{s}$ value in Appendix A). As shown in Appendix A, we found that MONSA obtains several equivalent excellent fits, as assessed by $\chi^{2}$—the mean deviation between the fit and the data—for most $f_{s}$ values probed. Hence, $\chi^{2}$ could not inform us of the loci of the solidus boundary directly. In order to extract this information, we included two additional constraints; the first required that the volume fraction of the liquidus obtained from the fit coincided with the expected liquidus volume fraction for that given $f_{s}$ value (Appendix A). The second constraint required that the molecular volume obtained from the fit (Appendix A) coincided exactly with the molecular volume measured by densitometry/calorimetry (Figure 3). Plots of the difference between the fit and the expected liquidus volume fractions as a function of the $f_{s}$ value for the lower temperature data, which included 9 °C, 11.9 °C and 14.9 °C, produced only one possible solution for each temperature, as shown in Appendix A. Further, as shown in Figure 4B, these values of $f_{s}$ satisfied the molecular volume obtained by calorimetry/densitometry. 

Although we successfully modeled the solidus as being composed of only gel-phase lipids in the low-temperature regime, we found that this model failed for the higher-temperature data, i.e., 18 °C, 22 °C and 24 °C. Instead, we modeled the solidus as composed of dDPPC in the gel phase and DLPC in some intermediate volume between the fluid state and the gel state using a sliding parameter α:(9)$$V_{DLPC_{s}}=\;\alpha V_{DLPC_{g}}+\left(1-\alpha \right)V_{DLPC_{f}}$$
where α=1 corresponds to the gel state and α=0 the fluid state. 

DLPC is known to have a broad melting transition occurring over nearly 10 degrees, as shown in Appendix A, and therefore, as the system approaches ≈ 27 °C, the molecular volume of DLPC could take a value between its fluid and gel state. Using this approach, we found that for the 18 °C data, MONSA fits satisfied the expected liquidus volume fraction condition for multiple values of α, including α=1, which corresponds to the volume of DLPC to its gel state (see Appendix A). However, only for one value of α (α=0.75) did the molecular volume obtained (using Appendix A) coincide with the measured value (as shown in Appendix A). In the case of 22 °C and 24 °C, setting the volume of DLPC to its gel-state value in the solidus gave no solution (see Appendix A). For these temperatures, the most consistent solution suggested a DLPC volume closer to its fluid state, with α=0.3 (see Appendix A). Alternative models that varied the volume of both lipids did not produce fits that satisfied all constraints either. Indeed, this suggests that this behavior is due to the unique melting profile of DLPC (see Appendix A). In Appendix A shows the family of molecular volumes obtained, highlighting the effect of a sliding value of α for the higher-temperature data and, in Appendix A.3B, their concomitant position marking the solidus boundary in the phase diagram. Figure 4B shows the molecular volume for the 1:1 mixture obtained by densitometry and calorimetry with the corresponding best-fit molecular volumes obtained from the SANS analysis, and in Figure 4C are their loci in the phase diagram. Figure 4A shows the corresponding best fits against the scattering data. 

SANS is a technique that detects nanoscale structures, and using real-space bead vesicle models allowed us to extract this information. Shown in Figure 5A are real-space bead vesicles whose structures produced the scattering curves shown in Figure 4A. Interestingly, the features observed are reminiscent of those observed in giant unilamellar vesicles made from a similar mixture, as shown in Appendix A. Due to the bead grid nature of the model, we were able to analyze the domain characteristics using the pair correlation function on the surface of the vesicle shell: g(R_arc length_) [39]. As shown in Figure 5B, we obtained an oscillatory behavior for the bead density variation of the liquidus phase in the membrane relative to the mean density. The wavelength, λ_domain_, of the domains was obtained by fitting the pair correlation function to an exponentially attenuated cosine function [40]. The wavelength of the oscillation in the pair correlation function is seen to increase with temperature, as highlighted by the arrow. The wavelength of the oscillation corresponds to a characteristic length—in this case, a characteristic domain size—as shown in Figure 5C. 

The correlation peak positions in the data (highlighted with black dots in Appendix A) also reflect a characteristic length through the relation 2π/Q_peak_. We find that both correlation lengths nearly overlap for *T*
≤18 ℃. For higher temperatures, the oscillation behavior of the pair correlation function is less obvious, and the fits show a more drastic increase in the characteristic length than the one derived from the position of the correlation peaks in the data as shown in Figure 5C. 

## 4. Discussion

As shown above, the data and analysis presented here had the necessary combination of techniques to unveil the nature of the two phases in the coexistence region for the DPPC:DLPC system. With calorimetry, the phase diagram shows that this system forms a homogenous liquid phase at high temperatures. At lower temperatures, a region of coexistence between a liquidus phase and a solidus phase emerges. The upper boundary of the coexistence region, also referred to as the liquidus boundary, was found to be well defined; that is, the boundary shows a smooth and continuous variation in the composition of the liquidus phase as a function of temperature for all compositions ($0\leq \chi_{DPPC}\leq 1$), as has been reported by Van Dijck et al. [3]. The lower or solidus boundary for $0\leq \chi_{DPPC}\leq 0.6$ was found to have a very slight variation in composition with temperature but not a horizontal isothermal behavior as reported by Van Dijck et al. [3]. For 0.6 > $\chi_{DPPC}\geq 0.7$, the calorimetry-derived phase diagram displays a gap or discontinuity in the solidus boundary. This behavior of the solidus boundary is also observed in the binary system DSPC:DMPC (distearoyl phosphocholine (DSPC) and dimyristoyl phosphatidylcholine (DMPC)) reported originally by Mabrey et al. [41] and others [42,43] and more recently by Losada-Perez et al. [44]. Because DSPC: DMPC and DPPC:DLPC are mixtures of saturated lipids that differ in tail length by four CH_2_ groups, it is not surprising to find that their phase diagrams have the same features. 

Continuity between the lower branch (*χ_dDPPC_* < 0.6) and the upper branch (*χ_dDPPC_* > 0.7) of the solidus boundary has been sought theoretically [3] and experimentally using nano-probes (Foster Resonance Energy Transfer) [28] as well as by fluorescence confocal microscopy using giant unilamellar vesicles [23], and they suggest that the lower branch (*χ_dDPPC_* < 0.6) of the solidus boundary continues smoothly—without a sharp turn where 0.6 < *χ_dDPPC_* < 0.7—to *χ_dDPPC_* = 1, thus not coinciding with the upper branch described by the calorimetry result. Interestingly, consistency between molecular volumes and SANS fits resulted in a solidus boundary that connects the lower and upper solidus branches (see Figure 4C). Knoll and collaborators, in their exploration of the DSPC:DMPC phase diagram using a similar combination of approaches, calorimetry and densitometry [27] and SANS [45], found a similar discontinuity region suggested by densitometry and had a near-constant composition solidus boundary connecting the lower and upper solidus branches as well. SANS, however, was applied differently than was presented here. In their case, they performed solvent contrast variation experiments (varying D_2_O/H_2_O ratios) for two mixtures with different initial compositions [45] from which they were able to deduce the solidus and liquidus boundaries for the system, albeit for only a small number of data points. Using this approach, they additionally observed that in the solidus region, below the solidus boundary, the solidus phase was composed of two solidus phases. From the molecular volume behavior for the 1:1 mixture, we could deduce that this also occurs in the DPPC:DLPC system. Although most phase diagrams with peritectic characteristics are shown with an isothermal (horizontal) solidus boundary branch, we found that in the DPPC:DLPC system, the lower branch of the solidus boundary (*χ_dDPPC_* < 0.6) is not horizontal but instead shows a gradually changing composition with temperature. Lozada-Perez et al. found similar behavior in the DSPC:DMPC system [44]. 

From the molecular volumes measured for the individual lipids as well as for the 1:1 mixture, we deduced that both DLPC and DPPC behaved gel-like in the solidus phase and liquid-like in the liquidus phase of the coexistence region. The combination of the molecular volume behavior for the 1:1 mixture and the SANS fits provided details on the nature of each phase; the liquidus phase was found to always consist of fluid-phase lipids while the solidus phase was found to be composed of gel-phase lipids only at low temperatures, and at higher temperatures (at and above 18 °C), DLPC was found to take intermediate volume states between its gel and fluid phase. To the best of our knowledge, this behavior had not been reported earlier. 

The binary mixture DPPC:DLPC, which is not an ideal mixture, is described by regular solution theory. We follow the thermodynamic approach of Van Dijk et al. [3] to obtain the phenomenological thermodynamic curves that reflect the boundaries of the phase diagram and are represented in Equation (10).
(10)$$T_{EGV}\left(\chi_{dDPPC}\right)=\frac{\frac{\Delta H_{DLPC}}{R}\left(1-\chi_{dDPPC}\right)+\frac{\Delta H_{dDPPC}}{R}\left(\chi_{dDPPC}\right)+\left(A^{L}-A^{S}\;\right)\left(\chi_{dDPPC}\right){\left(1-\chi_{dDPPC}\right)}^{n}n^{-n}{\left(n+1\right)}^{n+1}}{\frac{\Delta H_{DLPC}}{RT_{m-DLPC}}\left(1-\chi_{dDPPC}\right)+\frac{\Delta H_{dDPPC}}{RT_{m-dDPPC}}\left(\chi_{dDPPC}\right)}$$

Here, $\Delta H_{DLPC}$ and $\Delta H_{dDPPC}$ are the melting enthalpies for DLPC and dDPPC, respectively, and $T_{m-DLPC}$ and $T_{m-dDPPC}$ are their respective melting temperatures. In this equation, n represents a nonlinear exponent of the excess enthalpy and provides the necessary asymmetry between the solidus and liquidus boundaries. The parameters $A^{S}$ and $A^{L}$ describe the maximum excess enthalpy for solidus and liquidus, respectively. Equation (10) describes the temperature vs. composition curve for the case when the free energy of the solidus and liquidus is equal. On the phase diagram, this curve lies approximately mid-way between the solidus and liquidus boundaries. We found that values of n=2, $A^{L}\;$= 112 cal·mol^−1^ and $A^{S}\;$= 172 cal·mol^−1^ reasonably describe the boundaries as shown in the Figure 4C, which are not far from the values reported by Van Dijk et al. earlier (n=1.65, $A^{L}\;$= 75 cal·mol^−1^ and $A^{S}\;$= 250 cal·mol^−1^). These values indicate that the maximum excess enthalpy difference between the liquidus and solidus, proportional to the difference $A^{L}-A^{S}$, is much smaller in our analysis than that found in the interpretation of Van Dijk et al. The polynomial representing the excess enthalpy ($\left(\chi_{dDPPC}\right){\left(1-\chi_{dDPPC}\right)}^{n}$), on the other hand, only differs minimally between n=2 and n=1.65. 

SANS is a technique that can be used to extract structural information in addition to detecting phase separation in vesicles. Anghel et al. [18] studied the dDPPC:DLPC system above and below the miscibility transition using 30 nm in diameter vesicles to capture the characteristic peak from the domains by SANS. As in the present study, their goal was to extract structural information. They approached the analysis using analytical models such as round domains and stripes but were unable to show consistency between their analysis and the boundaries from the phase diagram reported by Van Dijk et al. Thus, using analytical shapes such as circular domains and stripes, which, according to GUV imaging (see References [22,23] and Appendix A), were unlikely, was an important limitation of their analysis. Using a bead vesicle as a real-space model to fit the SANS data from vesicles exhibiting phase separation allowed for a free-form analysis tool that yielded domains showing a phase (liquidus) evolving into a network of elongated dendrite-like features, as shown in Figure 5A, and which are similar to those found in GUVs (see References [22,23] and Appendix A). Notwithstanding, we also pursued using round domains to analyze the SANS data using a script in Igor^TM^, where we could generate round domain configurations (varying the number of domains but keeping the volume fractions constant). In the high-temperature regime (14.9 ℃≤
*T*
≤24 ℃), round domains produced larger intensity curves than the data suggested. Lower intensity is attained in the free-form approach by domain structures that “finger” out, as shown in Appendix A. For the lower-temperature regime (9 ℃≤
*T* ≤12 ℃), the congruency between the data and the calculated scattering for round domains was found to be significantly better. This congruency is partly explained by the ability to form dendrite structures with many small round domains (see Appendix A). In conclusion, given the robustness of the elongated features obtained through MONSA over repeated fits and the images of solidus and liquidus phases of the DPPC:DLPC system in GUVs and by comparing with what can be achieved with round domains, the real-space bead vesicle configurations obtained with MONSA appear physically reasonable.

It is known that this system exhibits a membrane thickness mismatch between the different domains [46]; that is, the solidus domains are expected to be thicker, due to a higher content of dDPPC, than the liquidus domains. In addition, it is known that membrane mismatch can drive domain size [17], where an increase in domain size is associated with an increase in membrane mismatch between domains. This latter result from Heberle et al. suggests that the decreasing characteristic domain size of the liquidus phase is correlated with a decrease in membrane mismatch between solidus and liquidus with decreasing temperature. This is reasonable since, as the temperature lowers, the composition of the solidus changes to include an increasing fraction of DLPC, which would result in the thinning of the solidus phase. 

## 5. Conclusions

In closing, we have shown that the use of complementary techniques, which in this case consisted of calorimetry, densitometry and SANS, was necessary to obtain a revised phase diagram for the DPPC:DLPC system. We find that the lower ($\chi_{DPPC}<0.6$) and higher ($\chi_{DPPC}>0.7$) solidus boundary branches connect, contrary to previous reports [3,23,28]. Consistency between these separate methods revealed that the solidus phase was composed of gel-phase lipids only at low temperatures, and at higher temperatures, DLPC was found to take intermediate states between its gel- and fluid-phase volumes. This behavior for DLPC was found to be consistent with its unusual gel-to-fluid transition temperature behavior [41]. In addition, we obtained domain structure information using a free-form real-space bead vesicle model and found the domains’ shapes to be consistent with a modulated phase [40], whose characteristic size decreases with decreasing temperature. This behavior correlates with a decreasing membrane mismatch between the solidus and liquidus phases due to a concomitant thinning of the solidus phase due to an increase in DLPC content and a thickening of the liquidus phase due to an increase in DPPC content [17]. 

## Figures and Tables

**Figure 1 membranes-13-00323-f001:**
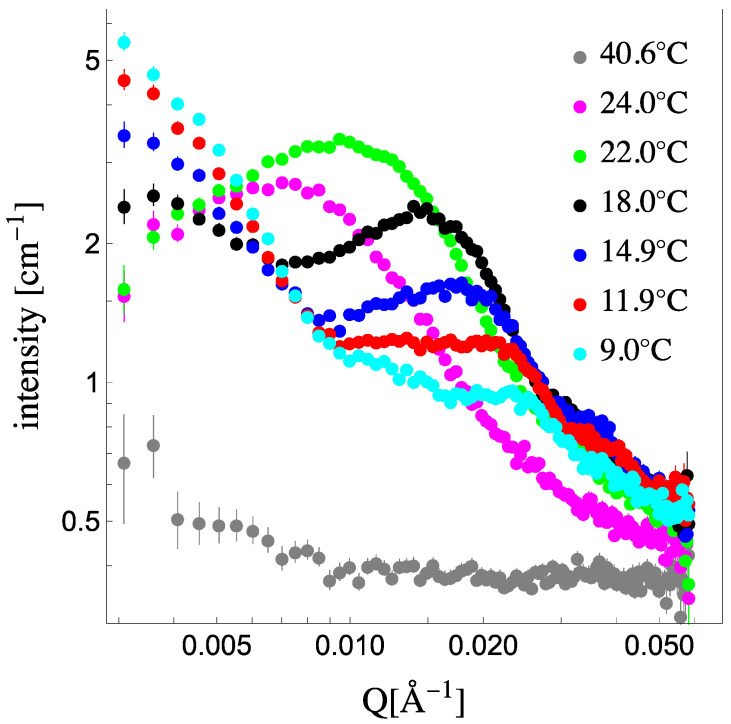
SANS spectra for 20 mg/mL, 50 nm in diameter small unilamellar vesicles of a 1:1 mixture of tail-deuterated DPPC (dDPPC) and DLPC at different temperatures. The solvent matches the mean SLD of the vesicles at high temperature (40.6 °C) when no domains are present. At lower temperatures, excess scattering appears due to the emergence of solidus and liquidus domains, which unequally partition deuterated and non-deuterated lipids, creating contrast between regions within the membrane as well as the solvent.

**Figure 2 membranes-13-00323-f002:**
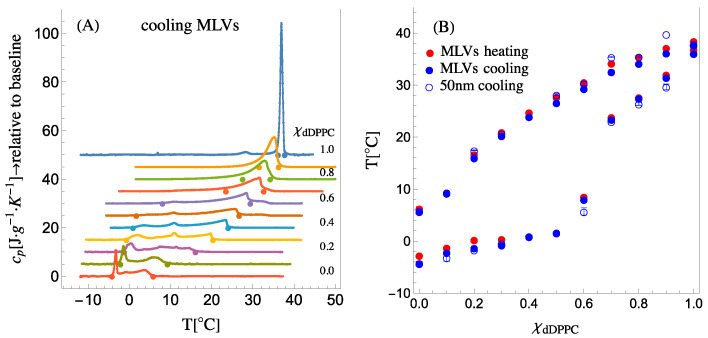
(**A**) Cooling calorimetry traces for MLVs for mixtures of dDPPC:DLPC. Large dots mark the onset and the completion temperatures of the transition. The dots trace the liquidus and the solidus boundaries. (**B**) The composition, $\chi_{dDPPC}$, vs. temperature, T, plot corresponding to the phase diagram for the dDPPC:DLPC system. The solid dots correspond to MLVs, blue for cooling and red for heating. The open blue circles correspond to 50 nm SUVs during cooling. Error bars correspond to at least two calorimetry scans (see Appendix A for details).

**Figure 3 membranes-13-00323-f003:**
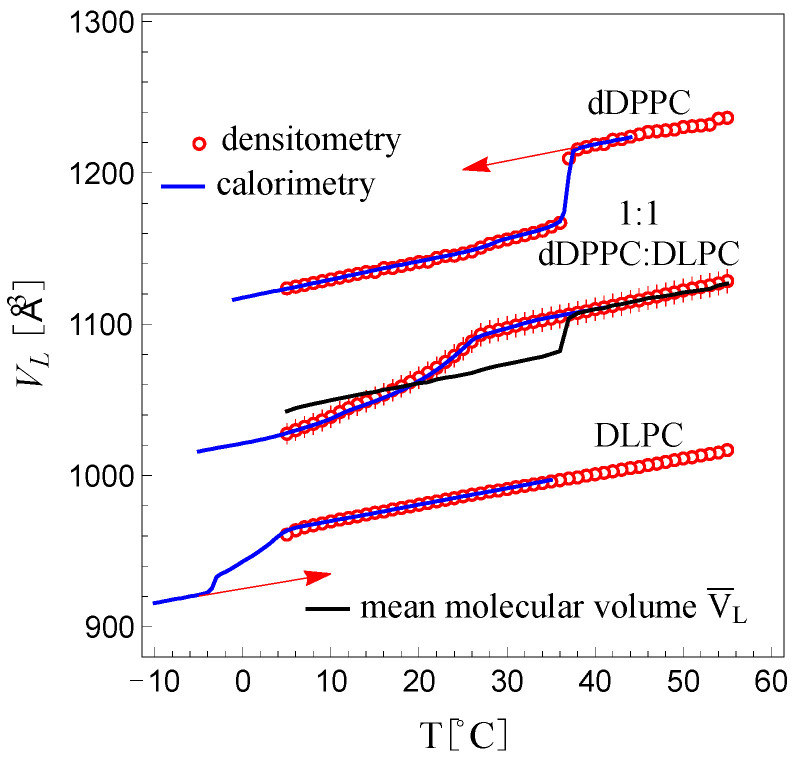
Molecular volumes, *V_L_*, for dDPPC, DLPC and the 1:1 mixture of dDPPC and DLPC obtained by densitometry (red circles) and by inversion of calorimetric traces (blue) according to Equation (3). The red arrows indicate corresponding *V_L_* extrapolations: see Appendix A for the resulting linear, temperature-dependent molecular volumes for dDPPC and DLPC above and below *T_m_*. The black trace shows the 1:1 molar additive mean volume obtained from the densitometry traces for dDPPC and DLPC. Error bars for *V_L_*, δ*V_L_*, were calculated using the error in weighing (±0.0001 g) and the error in the density (±0.0001 g/mL) and are shown in Appendix A; for dDPPC and DLPC, the error bars are approximately the size of the symbol (δ*V_L_* ≈ 3–4 Å), while for 1:1 dDPPC and DLPC, δ*V_L_* ≈ 7 Å.

**Figure 4 membranes-13-00323-f004:**
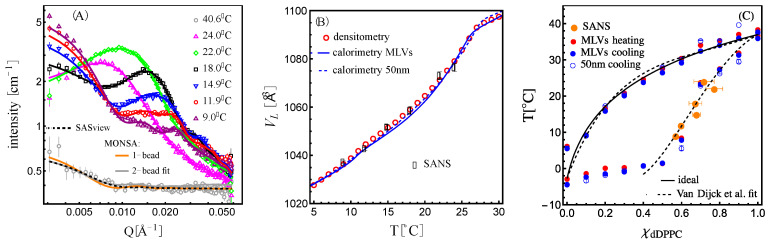
(**A**) SANS data for 1:1 dDPPC:DLPC 50 nm vesicles in contrast-matched solvent. For the 40.6 °C data, three curves are over imposed: a SASview [38] fit, which uses the analytical vesicle form factor, with both polydispersity and instrument smearing (dashed line); a MONSA calculation with all beads having the same ${\mathrm{SLD}}_{mean}$ (orange); and a fit with MONSA using two SLDs: ${\mathrm{SLD}}_{dDPPC_{f}}$ and ${\mathrm{SLD}}_{DLPC_{f}}$ (gray). The MONSA fits for data below the miscibility temperature use two SLDs: ${\mathrm{SLD}}_{s}$ and ${\mathrm{SLD}}_{l}$. The fits also satisfy the expected volume fraction for the solidus and liquidus phases (see Appendix A) as well as the molecular volume, given by densitometry and calorimetry (shown in (**B**)). (**B**) Molecular volumes derived from densitometry/calorimetry and those obtained from SANS fits. (**C**) Phase diagram shown in Figure 2 but including the solidus boundary extracted from the SANS fits constrained by the molecular volume—shown in (**B**)—and the expected solidus and liquidus volume fractions (Appendix A). Dashed lines delineate a fit to the solidus and liquidus boundary obtained using the approach by Van Dijck et al. [3]. The ideal liquidus boundary is shown with a black curve.

**Figure 5 membranes-13-00323-f005:**
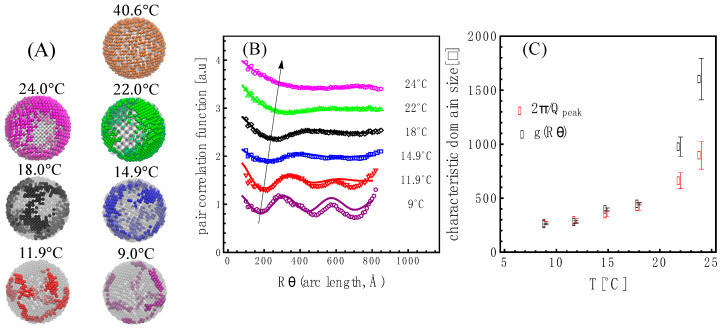
(**A**) Bead models as a function of temperature. At 40.6 °C, the two-color beads represent dDPPC—golden—and DLPC molecules—light gray—in the fluid state and where $\upsilon_{DLPC}=0.448$. At 24 °C and lower temperatures, in which the system is in the two-phase coexistence region, the colored beads represent the liquidus phase while the light gray beads correspond to the solidus phase. The corresponding scattering from these structures is shown in Figure 4A. The error on the volume fraction is ±0.002 as shown in Appendix A. Appendix A, lists the corresponding volume fraction of the liquidus obtained. (**B**) Pair correlation function, g(R_arc length_). The curves are offset for clarity. The curves correspond to fits with an exponentially attenuated cosine function. The arrow indicates the increase in wavelength. (**C**) The characteristic length obtained from fitting g(R_arc length_) as a function of temperature (black symbols). The characteristic length obtained from the scattering peak is plotted as well (red symbols).

## Data Availability

The data presented in this study are available in the Appendix A.

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
