# Peer review of "A Small-Angle Neutron Scattering, Calorimetry and Densitometry Study to Detect Phase Boundaries and Nanoscale Domain Structure in a Binary Lipid Mixture"

_membranes, 2023, doi:10.3390/membranes13030323_

Round 1

Reviewer 1 Report

The authors reported the analysis of phase separated lipid membrane by a small angle neutron scattering, calorimetry and densitometry. Their findings are interesting because their results contained difference compared to that of previous reported papers, and their methodology will give us more accurate insight about the phase separated lipid membranes.

I hope my following comments will be helpful for improvement of their manuscript.

(1)   Please do not forget to add the key words.

(2)   Line 171, abut this peak, nobody can understand what peak the authors referred.

(3)   Why did the authors select the 50 nm as size? In addition, how size did they prepare MLV? Please mention about them on their manuscript.

(4)   The sentence of “This behavior revealed that a fraction of DLPC is not in the fluid phase” is correct? This sentence is in contradiction with next sentences of “Hence, from these measurements, we inferred that the solidus must be comprised of only gel phase lipids while the liquidus contained only fluid phase lipids.”

(5)   Please clearly define the word, soliduis, liquidus, gel phase, fluidic phase. I cannot understand why the author use the different symbol, the index s or l, indicates solidus or liquidus, and ? or ?, indicates gel or fluid? It seems to be complicated.

(6)    Please make one space between numerical values and symbol.

Author Response

We would like to thank the reviewers for their comments and recommendations to our manuscript. Below our responses to their comments:

Reviewer 1:

(1) Please do not forget to add the key words.

We have added key words. Thank you.

(2) Line 171, abut this peak, nobody can understand what peak the authors referred.

Indeed, as the reviewer points out, the correlation peak is not vey obvious particularly for temperatures below 22ºC. The corresponding paragraph describing the data in figure 1 has been modified to address this comment. The new paragraph is highlighted in yellow in the manuscript (pages 5 and 6) and included below:

“Figure 1 shows the scattering from 1:1 dDPPC:DLPC 50nm in diameter vesicles in their contrast matched (CM) solvent (see section S2 for details on obtaining the CM condition for the system) above and below the miscibility temperature. At 40.6ºC, which is above the miscibility transition for this mixture, the scattering signal is flat, confirming that the membrane’s SLD has been matched to the solvent’s SLD. Upon lowering the temperature, below the miscibility transition, phase separation into solidus and liquidus domains ensues and their presence is clearly captured by a significant increase in the scattering intensity [14]. This emergent scattering, however, is not due to the vesicle form factor as it is contrast matched to the solvent, but due to the characteristics of the domains in the vesicles. In order to extract structure and composition information from these coexisting phases in vesicles, beyond just detecting their presence, requires that the collected scattering signal captures the peak resulting from the size of the domains present in the vesicles. Using 50nm vesicles allows the capture of this peak[17, 23], as it is clearly apparent at 24ºC and 22ºC. As temperature decreases, we observe the scattering signal evolve. Initially the scattering peak increases in intensity and shifts to higher Q values (from 24ºC to 22ºC), then low Q scattering increases while the scattering peak lowers in intensity while still shifting to higher Q values. At the lowest temperatures, the scattering curves have significant low Q scattering while the peak has become broad and shoulder-like in the high Q. This evolution in the scattering spectra suggests that the membranes’ domains are not only changing in lipid composition, size and number but the lipids’ SLD changing as well. Therefore, in order to quantitatively analyze the SANS data presented in figure 1, we need to constrain the analysis with the thermodynamic parameters of the system. The phase diagram provides the lipid composition of each phase and fraction of each phase [26] while the volumes of the lipids provide the SLD of each phase [27]”   

We additionally have added a figure (figure S10) in the SI that mark the maxima/inflection points of the characteristic peaks used to produce figure 5C.

(3a) Why did the authors select the 50 nm as size?

The added text in (2), we address the reason why we selected 50nm vesicles: the characteristic peak, which is due to the size of the domains, is fully captured in 50nm vesicles while in 100nm it is not: the peaks are incomplete or truncated. 

As a side a note, this size of vesicles (50nm) have also been used by Heberle at al. and Balbatov et al. (references 17 and 23) for the same reason: capturing the characteristic peak resulting from the domain size in the vesicles.

(3b) In addition, how size did they prepare MLV? Please mention about them on their manuscript.

We mention in the manuscript how we prepared the MLVs but have added more details (marked in yellow in the manuscript):

“MLVs were prepared by leaving the solution on a shaker overnight mixing in a 47°C oven until the resulting mixture became a homogenous suspension.”

Before this sentence we describe the mixing of lipid powders using chloroform and then drying in N2 followed by vacuum at 60ºC to remove the residual solvent before adding water.

(4) The sentence of “This behavior revealed that a fraction of DLPC is not in the fluid phase” is correct? This sentence is in contradiction with next sentences of “Hence, from these measurements, we inferred that the solidus must be comprised of only gel phase lipids while the liquidus contained only fluid phase lipids.”

In order to clarify this apparent contradiction, we re-wrote these sentences (and highlighted then in yellow in the text in page 7) as follows:

“This behavior revealed that a fraction of DLPC must be in the gel state. Hence, we inferred that the solidus phase must be only comprised of lipids in the gel state while the liquidus phase must only consists of lipids in the fluid state.”

(5) Please clearly define the word, solidus, liquidus, gel phase, fluidic phase. I cannot understand why the author use the different symbol, the index s or l, indicates solidus or liquidus, and ? or ?, indicates gel or fluid? It seems to be complicated.

In order to clarify why these different terminologies are used, we added the following sentences in the text (highlighted in yellow in page 9):

“The terminology difference between solidus and gel or liquidus and fluid is meant to distinguish between the state of a lipid mixture versus the state of a lipid species. As will be shown below, this distinction is necessary.”

We believe the distinction is necessary because it is consistent with the terminology used in the description of the phase diagrams of binary lipids mixtures but also because in this system, as it approaches the upper transition boundary (27C), the lipids in the solidus phase are not just lipids in the gel state: instead, we find that DLPC adopts a molecular volume that is in between its gel a fluid state. 

(6) Please make one space between numerical values and symbol.

We have, though if we missed some, the editorial office will hopefully help us out.

We thank the reviewer for the suggestions and feedback.

Author Response

We would like to thank the reviewers for their comments and recommendations to our manuscript. Below our responses to their comments:

Reviewer 2:

We have added the references suggested by the reviewer in addition to this reference:

Domains on a Sphere: Neutron Scattering, Models, and Mathematical Formalism

  1. N. P. Anghel, D. Bolmatov, J. Katsaras and J. Pencer

Chemistry and Physics of Lipids 2019 Vol. 222 Pages 47-50

These references where cited in the introduction and the text is highlighted in yellow. A sentence in page 2 was added (and highlighted in yellow) to refer to the work of Bolmatov et al. (https://pubs.acs.org/doi/abs/10.1021/acs.langmuir.9b01534) as follows:

“More recently, Bolmatov et al. developed a promising scattering model that uses a phenomenological free energy approach and whose fitting parameters could in principle identify domain configurations having diverse shapes; however, its implementation is still ongoing[20].”

We thank the reviewer for alerting us to these works which was certainly an oversight on our part!